# Development of a robust protocol for the characterization of the pulmonary microbiota

Nathan Dumont-Leblond [1,2], Marc Veillette[1], Christine Racine[1], Philippe Joubert[1,3] & Caroline Duchaine [1,2 ✉]

The lack of methodological standardization diminishes the validity of results obtained and the conclusions drawn when studying the lung microbiota. We report the validation of a complete 16S rRNA gene amplicon sequencing workflow, from patient recruitment to bioinformatics, tailored to the constrains of the pulmonary environment. We minimize the impact of contaminants and establish negative controls to track and account for them at every step. Enzymatic and mechanical homogenization combined to commercially available extraction kits allow for a fast and reliable extraction of bacterial DNA. The DNA extraction kits have a significant impact on the bacterial composition of the controls. The bacterial signatures of extracted cancerous and healthy human tissues from 5 patients are highly distinguishable from methodological controls. Our work expands our understanding of low microbial burdened environments analysis. This article is to be a starting point towards methodological standardization and the implementation of proper sampling procedures in the study of lung microbiota.

[1] Centre de Recherche de l'Institut Universitaire de Cardiologie et de Pneumologie de Québec, Quebec City, QC, Canada. [2] Département de Biochimie, de Microbiologie et de Bio-Informatique, Faculté des Sciences et de Génie, Université Laval, Quebec City, QC, Canada. [3] Département de Biologie Moléculaire, Biochimie Médicale et Pathologie, Université Laval, Quebec City, QC, Canada. ✉email: Caroline.Duchaine@bcm.ulaval.ca

The modification of the lung microbiota has been linked to many pulmonary pathologies, such as the chronic obstructive pulmonary disease (COPD)[1,2], asthma[3], idiopathic pulmonary fibrosis (IPF), and cystic fibrosis (CF)[4]. This pulmonary microbiota shift could also have an important impact on human health considering its influence in other body regions[5,6]. Therefore, the characterization and study of the pulmonary microbiota are of high importance.

Bacteria can impact carcinogenesis, the evolution of cancer, and the outcome of treatments of pancreatic and bowel cancers in mice[7–9]. A few studies aimed at analyzing a similar effect of lung microbiota on pulmonary cancers[10–13]. Recently, Jin et al. observed a promoting effect of commensal bacteria on lung cancer development in mice[14]. Such a distinct effect as yet to be confirmed in humans. Until now, human pulmonary microbiota studies integrating next-generation sequencing methodologies (NGS) have all used different protocols for tissue collection, nucleic acids extraction, and bioinformatics analyses, which limit the conclusions that can be drawn from the current literature. Therefore, the development of an accurate and standardized method for the characterization of the lung microbiota seems mandatory.

DNA extraction methods have a noticeable impact on the microbial community detected[15–17]. Since a wide variety of inhibitors (ions, polysaccharides, etc.) can reduce the efficiency of DNA extraction, protocols should be optimized to the sample matrix[18]. Besides, the resistance to lysis of some bacteria, such as Gram-positive bacteria[19], may reduce our ability to detect them, creating a bias toward more easily lysed genera. The microbial biomass in the lung is low compared to what is found in the digestive system[20]. Bacterial profiling of such low-density community is therefore more prone to biases induced by contaminants and method selection[21]. In addition, commercial DNA extraction kits may carry a substantial number of bacterial contaminants[22]. Precautions are required to ensure that detected microorganisms are not incorporated by the experimental method[23]. The implementation of strict contaminant management strategy is necessary[24].

The goal of our study was to validate multiple key aspects of a complete pipeline, from sampling to result analysis, that is designed and adapted to the study of the intratumoral lung microbiota. It is intended to be a first step toward achieving methodology standardization in lung microbiota studies. Patient selection guidelines and sample collection and processing methods were developed. Three different commercially available DNA extraction kits were tested and the QIAamp® DNA Blood Maxi Kit was identified as the most appropriate based on efficiency of bacterial recovery in lung tissues. The type of DNA extraction kit used influenced the nature of the contaminants detected. A close attention was given to the influence of methodological contaminants on the detected microbiota. The introduction of a single negative control for every patient, combined with appropriate bioinformatics processing, provides concrete ways to ensure a reliable description of the lung microbiota.

## Results

The validation of the proposed methodology is presented in two steps. First, bacterial communities of known compositions spiked at biological concentrations in lung tissues were used to quantify the possible influence of the pulmonary matrix on the DNA extraction and sequencing processes (Supplementary Fig. 1). Then, the base microbiota of lung tumoral and adjacent healthy tissue was examined to select the most appropriate version of the protocol, depending on the DNA extraction kit used (Blood,

Microbial, Powersoil) (Supplementary Fig. 2). Supplementary Fig. 3 illustrates the final protocol.

**Detection of spiked bacterial community**. Two types of bacterial community (20 species) were used: Whole-cell communities, comprised of live bacteria with intact cellular membranes, and genomic communities, containing only purified genomic DNA. These two communities were spiked into homogenized lung tissue or purified total lung DNA to assess the efficiency of recuperation of the method and better understand the influence of the abundance of human DNA in the samples on the 16S gene sequencing detection method selected. Considering the lack of in-depth lung microbiota characterization and its low biomass, mock communities were also used to provide a base line for expected recovered species and ensure that the detected microbiota was not only artifacts. Supplementary Fig. 1 provides a methodological overview of this section. Three technical replicates were performed for each version of these tests. Whole-cell bacterial communities were efficiently detected in the spiked homogenized tissues. The pipeline version using the Blood, the Microbial, and the Powersoil kits detected 90.7%, 100%, and 88.9% of the bacteria genera added, respectively. *Cutibacterium acnes* was not detected in any of the three replicates using the Blood kit and *Bacteroides vulgatus* was detected in only one of these replicates. *Bifidobacterium adolescentis* was detected once and *Deinococcus radiodurans*, *Clostridium beijerinckii*, *Helicobacter pylori*, *Lactobacillus gasseri* were found in two of the replicates using the Powersoil kit. In addition, non-spiked tissue homogenates confirmed that the genera were not naturally present in the tissue before the addition of the mock community. Only the genera *Acinetobacter* and *Staphylococcus* were detected in those control tissues. All the genera from the genomic mock-community ($n = 20$) spiked in purified pulmonary tissue DNA were detected by sequencing.

**Detection of the human microbiota**. Once the capability of bacterial detection in lung matrix was established using mock-communities, the detection of the base-level microbiota in lung tissues was attempted to compare the efficiency of recuperation of the methods using the three different DNA extraction kits. Supplementary Fig. 2 provides a schematic representation of the methodology. Briefly, cancerous and healthy tissues of five patients were collected, homogenized, and split into the three different DNA extraction protocols to allow comparisons. The performance of three common kits, namely the Blood, the Microbial, and the Powersoil kits, was examined. DNA yield and purity, alpha diversity of the tissues, and the relationship between the bacterial signature of controls and tissues samples were compared.

**DNA yields and purity obtained with the different extraction kits**. The yield and purity of DNA was evaluated using UV spectrometry. The Blood kit extracted a greater quantity (paired Wilcoxon signed-rank, $p = 0.062$) of total DNA for an equivalent quantity of tissue than the Microbial and Powersoil kits for cancerous tissues. It also performed better (paired Wilcoxon signed-rank, $p = 0.062$) than the Powersoil kit for healthy tissues. In addition, the microbial kit extracted more DNA per gram of tissue than the Powersoil kit, independently of the tissue type (paired Wilcoxon signed-rank, $p = 0.062$) (Supplementary Fig. 5). No significant differences were observed between the three extraction kits used on cancerous and healthy tissues for the 260 nm/280 nm ratios, with ratios ranging from 1.51 to 2.19 and 1.5 to 2.4, respectively (Supplementary Fig. 6). As for 260 nm/230 nm ratios, the cancerous tissues extracted with the Blood kit led to

significantly higher purity ratios (paired Wilcoxon signed-rank, $p = 0.031$) (Supplementary Fig. 7).

**Alpha diversity.** The Alpha diversity is useful to capture the number and distribution of bacterial taxa inside each sample. A higher diversity is usually preferred as it may be a good indicator of better bacterial recuperation. The Shannon alpha diversity tended to be higher using the Blood than any of the other extraction kit for both tissue types and is statistically different to the two other extraction kits with cancerous tissue (double-sided paired $t$-test, $p = 0.033$ and 0.016). No significant difference was detected among the Microbial and Powersoil kits (double-sided paired $t$-test, $p$-values = 0.52 and 0.96) (Fig. 1).

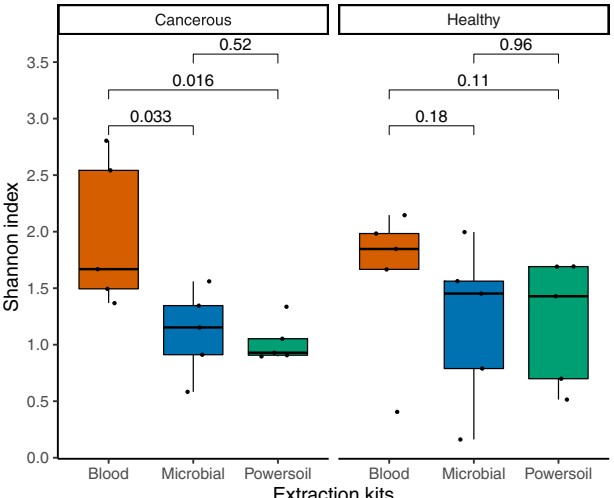

**Fig. 1 Shannon's alpha diversity of the tissue samples by type of tissue and extraction kit.** Double-sided paired sample $t$-tests were performed to account for the patient variable. The boxes and bars display the data range, quartiles, and median; $n = 5$ pairs of tissues (cancerous and healthy) from five different patients extracted by three distinct methods (total of 10 tissues, total of 30 extracts).

**Correlation to controls and removal of contaminating operational taxonomic units.** In order to assess the similarity in bacterial profiles between controls and sample (same patient and DNA extraction technique), Pearson's correlation coefficients were computed. A correlation value closer to 1 or −1 indicates a strong link between the two profiles and a coefficient of zero indicates the absence of correlation. The less correlated the samples are to the control, the more successful the bacterial DNA extraction from the tissue was and the less the contaminating bacterial load of the method impacted the final results. Both the Blood and Microbial kits had coefficients significantly closer to zero than the Powersoil (paired Wilcoxon signed-rank, $p = 0.027$, 0.0039) (Fig. 2a).

The percentages of remaining operational taxonomic units (OTUs) were computed by dividing the number of observed OTUs before and after the removal of the OTUs present in the corresponding control according to the developed bioinformatics pipeline. A result closer to 100% indicates a bigger difference in bacterial profiles between the tissue samples and the control and a reduced impact of the contaminating bacterial DNA carried by the extraction technic. The samples extracted with the Blood kit retained a significantly larger proportion of OTUs (88.46–96.69%) than the Powersoil kit (paired Wilcoxon signed-rank, $p = 0.0039$). No difference could be observed between the Microbial and Powersoil or Blood kits (paired Wilcoxon signed-rank, $p$-value = 0.28, 0.11) (Fig. 2b).

**Methodological core microbiota.** In order to assess the microbial contaminant burden of the method, the bacterial profiles of the controls were examined. The extraction kit used significantly explained a small portion of the variation observed in beta diversity between controls, both in weighted ($p = 0.019$, $R2 = 0.14$) and unweighted ($p = 0.018$, $R2 = 0.096$) metrics (Fig. 3a, b). The computation of core microbiota using ampvis2 revealed 10 OTUs shared by the controls from each version of the pipeline using the three different DNA extraction kits (Fig. 4). These were identified to be from the following genera: *Brevundimonas*, *Cutibacterium*, *Micrococcus*, *Nitrobacter*, *Ralstonia*, *Staphylococcus*, and *Xanthobacteraceae_unclassifed* (Fig. 5a). A total of 29,

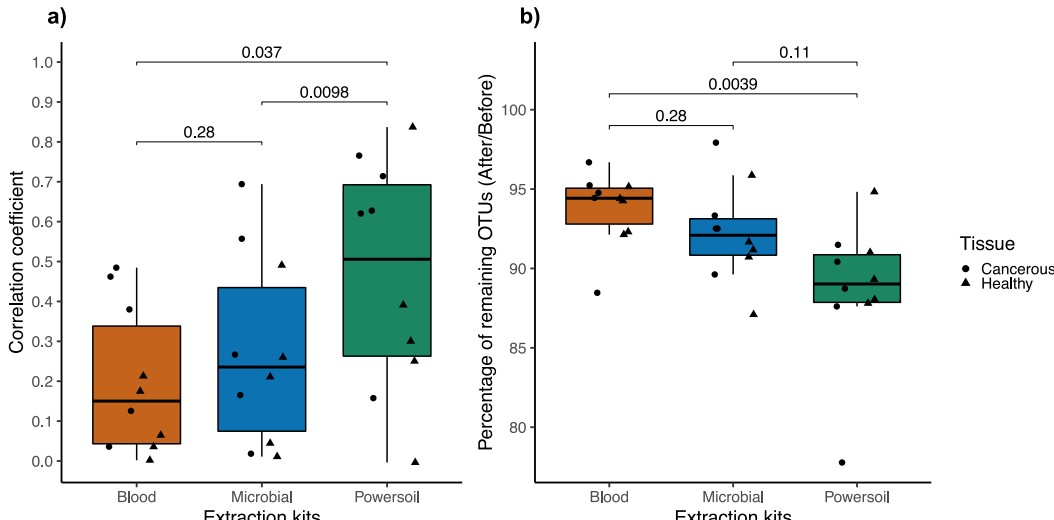

**Fig. 2 Comparison of the relationship between controls and tissue samples extracted with the three different methods. a** Pearson's correlation coefficient between tissue samples and corresponding control. A coefficient closer to 1 or −1 illustrates a strong relationship between the bacterial profile of the sample and the control. **b** Percentage of remaining OTUs after the removal of controls. A percentage closer to 100% shows that no OTUs present in the tissue samples had to be removed due to their presence in the controls. The boxes and bar display the data range, quartiles, and median. Paired Wilcoxon signed-rank tests were performed; $n = 5$ pairs of tissues (cancerous and healthy) from five different patients extracted by three distinct methods (total of 10 tissues, total of 30 extracts). Circles represent cancerous tissues and triangles healthy tissues.

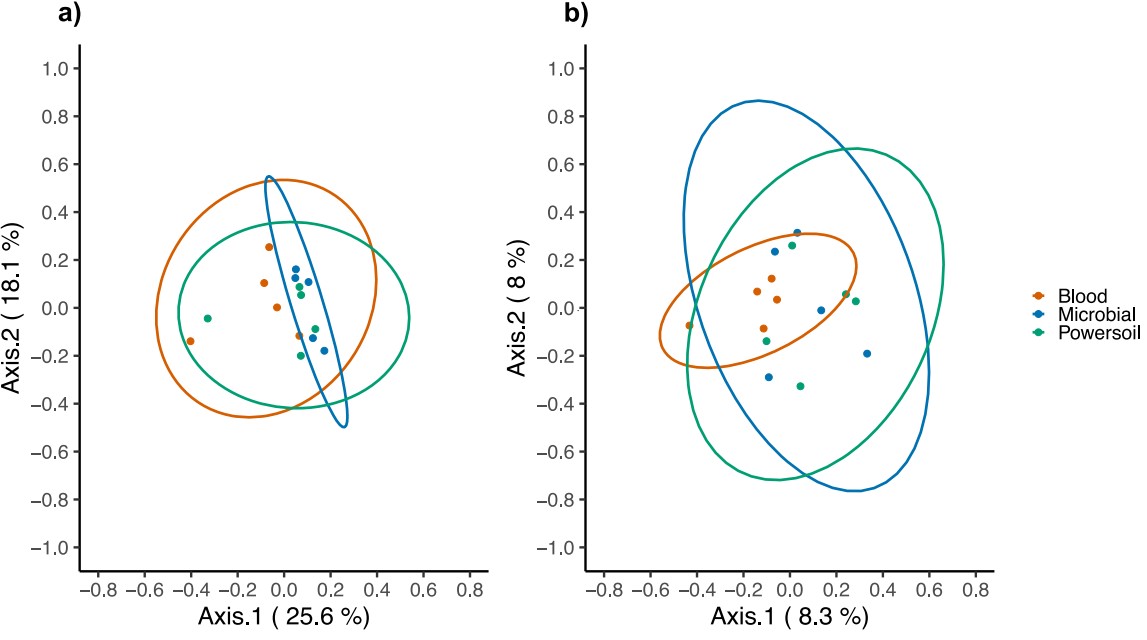

**Fig. 3 Principal coordinates analysis of the extraction controls by extraction kits. a** Based on Weighted Bray-Curtis Distances. **b** Unweighted Bray-Curtis Distances. Data ellipses were computed on multivariate *t*-distribution at 95%. The colors of the ellipses represent the three different DNA extraction methods used; *n* = 5 negative controls per method (total of 15).

34, and 25 different OTUs were detected only when using the Blood, Powersoil, or the Microbial kits, respectively. Their taxonomy was highly variable (Fig. 5b).

## Discussion

Many precautions should be taken to limit the modification of the commensal communities studied and the increase of inter-individual variation not attributable to the experimental variables. The following factors can influence the human microbiota and should be considered when designing studies targeting the lung microbiota: the administration of antibiotics or neoadjuvant[25–28], the size of the lesion, the type of surgical procedure, the type of pulmonary pathology under study, and living habits of patients (e.g., smoking status, physical exercise, buccal hygiene, alcohol consumption)[29–34].

A more exhaustive list of concomitant factors was pointed out by Carney et al.[35]. However, as the different fields of microbiota studies expand, it is likely that additional variables that can alter its composition will be uncovered. The molecular tools currently used to analyze the human microbiota do not have the power to discriminate the impact of that many factors over the microbial profiles. Whenever possible, patients selected for lung microbiota studies should be extensively screened so that they can be as similar as possible. Longitudinal studies could also minimize the impact of those variables, as the same patient, with similar concomitant factors through the study, would be compared to himself overtime.

Tissue management steps should consider the contamination possibilities. In addition to the selection of a less contamination-prone procedure, such as thoracoscopic lobectomy, the manipulations and the instrument used in subsampling the excised organ should be taken into account. A combination of bleach and humid heat was chosen to sterilize the instruments used to sample the cancerous and healthy tissue as it was considered the most easily accessible method. The use of humid heat itself (autoclave) lacks the power to completely neutralize bacterial genomic DNA in solutions and on surfaces[36]. On the other hand, the utilization of bleach, or a chlorinated detergent, leads to the

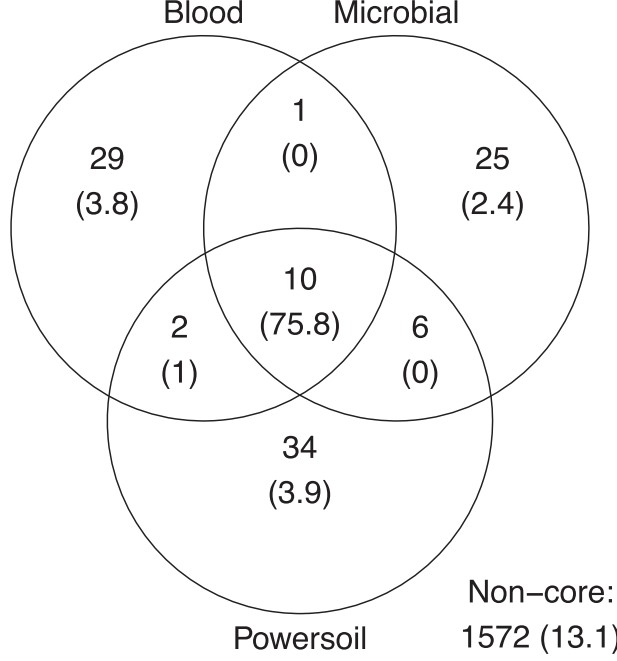

**Fig. 4 Venn diagram of the core microbiota in the methodological controls.** The numbers of OTUs with at least 0.001% of relative abundance in two of the five control samples are displayed in each part of the diagram. The numbers in parentheses represent the sum of the average relative abundance of the OTUs in the controls. Therefore, they may not add up to 100% for each category. The non-core statistic represents the total relative abundance or number of OTUs observed in at least 0.001% relative abundance but that could not be found in at least two samples of the sample version of the pipeline.

complete degradation of contaminating DNA on surfaces, such as benches and instruments[37,38], but requires rinsing to avoid corrosion. Hence, combining both methods, soaking the instruments in bleach 1.6% for 10 min before rinsing with distilled water and

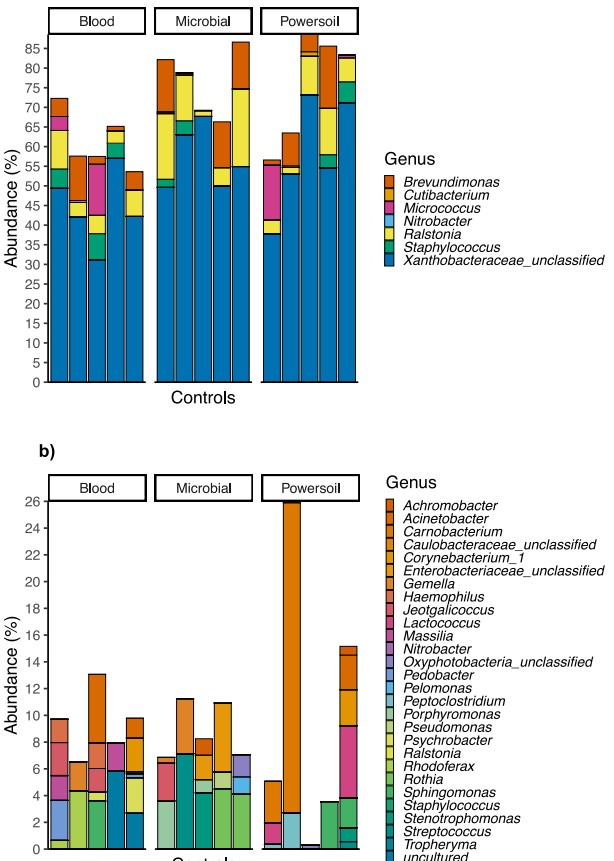

**Fig. 5 Distribution of the relative abundance and taxonomic identification of OTUs found in the controls for each version of the pipeline. a** Core microbiota of the methodological controls. Average abundances of OTUs present in at least 40% of controls for every version of the pipeline. **b** OTUs found only in the controls extracted with one or the other extraction kit. Average abundance of OTUs present in at least 40% of the controls from one of the versions of the pipeline, but not the others (different DNA extraction kits).

autoclaving in a sterilization pouches, ensures a minimal amount of DNA has to be degraded by moist heat. The rest of the single-use equipment used was commercially sterilized with ionizing radiation.

Healthy lung tissue was subsampled from the pulmonary lobe containing the tumor to ensure that the developed method could be used on a variety of lung tissue samples. It could also act as a control of non-pathologic microbiota to allow comparisons of cancerous and non-cancerous samples within the same subject, hence minimizing the impact on inter-individual microbiota variations. In fact, Riquelme et al. found that the gut microbiota has the capacity to specifically colonize pancreatic tissue[8]. Correspondingly, the use of adjacent pulmonary tissue to the tumor could help get better insights at a specific colonization of the tumor by lung bacteria. A 5 cm distance between the tumor and the healthy sample was ensured to minimize the potential effect of increased inflammation surrounding the tumor. Furthermore, the lung microbiota composition seems to vary dependently on the position and depth of the respiratory tract, even inside a same lobe[39]. The healthy tissue was collected in the same tierce of pulmonary depth (Supplementary Fig. 4) in an attempt to sample a microbial community that it would be as representative of non-pathologic microbiota in the tumoral region as possible.

The homogenization of frozen and thawed pulmonary tissues was attempted and was unsuccessful, both with the use of only a 2.8 mm tungsten bead in the Retsch – MM301 mixer mill (30 beats/s, 20 min) or of the Fisherbrand 150 homogenizer with plastic probes (Fisher scientific, Pittsburg). The elasticity of the tissue or its frozen state make the mass nearly unbreakable. The use of the Liberase™ TM enzymatic cocktail (collagenase I & II, thermolysin) prior to the mechanical homogenization proved successful and a homogeneous suspension was obtained using the two-step homogenization protocol (Supplementary Fig. 3). Multiple ratios of liquid to mass of tissue were tested and 3 mL/g was found optimal, as it facilitates the homogenization without overly diluting to sample. A similar ratio of liquid to tissue was used in breast tissue microbiota study[40]. The samples were first thawed at 4 °C to reduce potential growth or degradation of microorganisms. The digestion was performed directly in the 50 mL collection tube to limit the tissue manipulation and ensure possible contaminant tracking.

Our team was also unable to replicate the results obtain by Yu et al. on larger tissue samples using 0.2 mg/mL of Proteinase K for 24 h[13]. The samples remained firm and turned brown. Using the Liberase™ cocktail enabled a much faster digestion (75 min) and broke down specifically the lung component responsible for its elasticity, the collagen.

Three commercially available DNA extraction kits were tested. They were selected for their previous successful use in the study of pulmonary or gut microbiota and their intended application as described by the manufacturer. The extraction kits were first tested on homogenized lung tissue spiked with whole-cell bacterial community to assess the efficiency of DNA extraction and recuperation of the commercial kits. The three kits were able to recover more than 88% of the genera added to the samples. All the genera that were not detected by the Microbial and Powersoil (*Cutibacterium acnes, Bacteroides vulgatus, Bifidobacterium adolescentis, D. radiodurans, Clostridium beijerinckii, L. gasseri*), with the exception of *H. pylori*, were Gram-positive bacteria. This type of bacteria has been reported to require more aggressive extraction methods to break their tougher cell walls[19]. However, the bacterial community did not go through the enzymatic and physical homogenization that usually takes place before DNA extraction since we needed to obtain a homogenous tissue sample that could be processed with or without spiked bacteria. These hard to lyse Gram-positive bacteria could have been fragilized by these processes, rendering them easier to break down during the extraction protocol. Furthermore, the detection of the artificially incorporated bacteria does not account for the natural physical association that may occur between the human tissue and microbial cells. Nonetheless, these high percentages of recovery were promising and lead us to continue with the characterization of the extraction kits in a real-life context, meaning the analysis of the base-level microbiota in pulmonary samples collected and processed through the entire pipeline.

Every measurement of the efficiency of extraction, including DNA yield (Supplementary Fig. 5), DNA purity (Supplementary Figs. 6 and 7), and alpha diversity (Fig. 1), pointed in the same direction. In fact, they all showed that the Blood extraction kit was the best option out of the three kits. Therefore, using the Blood kit is recommended as one of the pieces of a complete study design. Additionally, the presence of a high concentration of host DNA in tissue samples might tend to saturate the purification column, which could reduce to amount of bacterial DNA recovered. The superior DNA binding capacity of the affinity column of the Blood kit compared to the two others could explain its better performance and its higher yield in most cases. The samples extracted with the Blood kit were also associated with higher alpha diversity (Shannon index). Therefore, this extraction

method was able to recover a higher number of different bacterial organisms (richness) and proportionality in the different OTUs (evenness). The absence of PCR inhibitors and a higher recuperation rate of bacterial DNA in the Blood extracted samples could have led to a more proper amplification in the sequencing process and to the recuperation of very low abundance bacterial DNA in the extraction eluate. For further research, it is advised to take the additional precaution of working under a biosafety cabinet or in the sterile field when analyzing the microbiota of lung tissues to reduce the risk of incorporation of airborne contaminants.

The Illumina Miseq sequencing platform with the use of dual-index strategy has become the dominant technology used in microbial ecology studies for its cost efficiency, low error rate, and user-friendliness[41–43]. Most studies interested in the pulmonary microbiota have also used this technology[11,13,14]. The sequencing of the 16S rRNA gene amplicon was favored over a shotgun sequencing method because of the overwhelming quantity of human DNA joining bacterial genomes in the pulmonary tissue. The 16S rRNA gene is the most used marker of bacterial identification. No consensus has been reached on the selection of the 16S rRNA gene variable region (V) to sequence for human microbiota[18,44]. However, it should be kept consistent across studies to allow comparisons. Targeting the V3–V4 regions was suggested using the universal primers developed by Klindworth et al.[45]. Several microbiota studies, including lung microbiota, have also used these regions[7,13,46–48].

In the context of this study, genomic mock-community was spiked in DNA extracted from the pulmonary tissue at a biological meaningful concentration. Every genus added to the samples was successfully detected. Consequently, the high ratio of human DNA to bacterial DNA did not interfere with the amplification and detection steps of the sequencing procedure. The sequencing method in place seems adequate for its application in the characterization of pulmonary microbiota.

Contaminating bacteria or DNA can have an important impact of the microbial profile observed in very low biomass samples such as pulmonary tissue[23]. Consequentially, in addition to proper protocol selection, methodological design that attempts to follow, detect, and account for contamination was proposed. Its main features include the incorporation of a single negative control that monitors the incorporation of contaminants at every step of the experimental method (Supplementary Fig. 3). Since every step of the protocol prior to the extraction is meant to be executed in a single tube and only by the addition of reagents, it is possible to carry and detect the contaminants introduced throughout the procedure. On the contrary, microbiota study methodologies usually dictate for the incorporation of multiple controls at every step of the procedure (e.g. DNA extraction kit, PCR controls, etc.)[18]. Although more informative as to which step leads to contamination, it makes data analysis harder since the presence of contamination in the multiple controls cannot by added.

No bioinformatics standard operating procedure is available and what should be done with controls sequencing data is still under debate[18]. Some research groups tried to use a neutral community model[49], additional qPCR data[50], amplicon DNA yield, or prevalence algorithms[51] to assess the influence of methodological contaminants. The removal of every bacterial OTU found in controls from the samples is often not appropriate as these OTUs might also be naturally present in the samples[22]. We propose using relative abundance ratio between samples and controls to remove contaminating OTUs. Since controls have much lower richness than extracted lung samples and that the total number of reads (sequencing depth) is distributed across every OTU, the relative abundance of reads for each OTU tend to be much higher in the control than the same OTU in samples. Therefore, if the relative abundance of an OTU is greatly superior in the sample than in the control, it is reasonable to think that the same OTU was also in the sample in a substantial quantity. To ensure that OTUs that were present in very low absolute abundance (e.g., from only 1–2 reads) do not lead to the removal of the highly abundant corresponding OTU in samples, only the OTUs with a ratio of 1000 (relative abundance of sample/relative abundance of sample) were kept. The rest of the OTUs found in controls were completely removed from the related samples, since the influence of contaminating DNA could not be differentiated from the pulmonary microbiota. This method would theoretically tolerate no more than 20 reads (0.1%) before removing the entire OTU from the sample if only one OTU was present in the samples (20,000 reads, 100%). The use of relative abundance helps reduce the absolute abundance bias induced by the divergence in sequencing depth. The OTUs were removed from both tissues at the same time or not at all to avoid adding artificial intraindividual variation. The authors acknowledge that the proposed contaminant management method does not have the in-dept validation of other methods, such as described by Davis et al. with the decontam package[51]. However, it does not share its limitations regarding the lack of consideration for OTU abundance and need of high number of controls to ensure sensitivity while using prevalence-based detection. Further research focused on the development of statistical methods to detect contaminant OTUs in the cases of lung microbiota is needed. This work is to be a starting point toward methodological standardization and its modular nature makes the bioinformatic contaminant management method proposed here interchangeable once a more robust one is uncovered.

Pearson's correlation tests were performed on the number of reads per OTU between the samples and their respective controls. Although these values were not normally distributed (Shapiro-Wilk, $p < 0.05$) and were zero inflated, the Pearson's test was still used, as it was found serviceable in these conditions by Huson et al.[52] and did not have an appropriate alternative. The correlation of the sample extracted with the Blood and Microbial kits to their controls was significantly lower than the one extracted with the Powersoil ($p = 0.027$, $0.0039$), which could indicate a higher level of contaminants in this last kit. The outperforming results of the Blood kit in the terms of DNA yield, purity, and alpha diversity corroborate its low correlation of its controls. In fact, the more bacterial DNA recovered from the sample, the lower the impact of contaminant present on the bacterial profile.

The extraction kit had a significant impact on the composition of the bacterial profiles found in the methodological controls, taking into account reads abundance ($p = 0.019$, $R2 = 0.14374$) or not ($p = 0.0184$, $R2 = 0.09582$). It is therefore a major contributor in the incorporation of contaminants in the samples. Ten genera were identified as "core" in the controls of every variation of the pipeline. Other steps of the protocol, such as the enzymatic homogenization and the sequencing method, were shared by all the pipelines and could have led to these similarities. The experimental design does not allow us to identify the origin of the contaminants.

The method described here shows some limitations. For instance, it might not be suitable for the use of culture method for the characterization and identification of microorganisms. The presence of thermolysin in the enzymatic homogenization cocktail could inadvertently reduce the recuperation rate of live organisms. Furthermore, the use of a 16S rRNA gene amplicon sequencing approach rules out the possibility of identification of fungal and viral microorganisms. The modification of the sequencing techniques could allow more versatility in targeted organisms.

Although great care was put in reducing the incorporation of bacterial contaminants in lung tissue samples, the methodological biases were not specifically measured. This preferential recovering and detection of some bacterial member over others is still of concern. The resources available to correct these biases are still very scarce, but McLaren et al. offer great evidence of its importance and concrete attempts[53].

This method allows the identification of the bacterial members of the lung community, but not their functionalities. Therefore, it could be interesting to develop a metatranscriptomics methodology for the pulmonary environment to get better insight at the transcribed microbial genes, as performed in the gut microbiota field of research[54].

Here, a comprehensive methodological pipeline for the study of lung cancer microbiota is proposed (Supplementary Fig. 3). A protocol of tissue collection and sample treatment that minimizes the risk of contamination was validated, a DNA extraction technique was adapted, and a bioinformatics pipeline to account for detected contaminants was designed. It is a first step toward protocol standardization. Although the perfect method does not exist, taking the appropriate precautions and being aware of its limitations ensure that appropriate conclusions are drawn and that the pulmonary microbiota field grows in a conscientious and reliable way.

## Methods
Supplementary Figs. 1 and 2 describe the protocol used to optimize the method. The recruitment and tissue collection steps are adapted to the clinical workflow in thoracic surgery and pathology at Institut Universitaire de Cardiologie et de Pneumologie de Québec (IUCPQ) in order to minimize the impact of the implementation.

**Patient selection**. Patients undergoing lung resection for pulmonary cancer between September 2018 and November 2019 were recruited through the IUCPQ Biobank. In order to be eligible for the study, patients had to meet the following requirements: (1) absence of antibiotic treatments 3 months prior the surgery; (2) absence of neoadjuvant therapy; (3) tumor larger than 2.0 cm; (4) lobectomy; and (5) diagnosis of either lung adenocarcinoma or squamous cell carcinoma. A total of five patients were enrolled for the project. The clinical details of the patients and specimens are summarized in the Supplementary Table 1. The project was approved by the IUCPQ ethic committee (project #1200). Ethical regulations were followed and informed consent was obtained.

**Sampling**. In patients undergoing lung resection for cancer, the excised organ can be sampled and used to describe pulmonary microbiota. However, the surgical and subsampling process can be susceptible to the incorporation of contaminants. To reduce these risks, lobectomies were performed using laparoscopy and the pulmonary lobes were kept in the sterile bag used to remove the organ from the patient until gross examination in pathology. The pathologist would lay the pulmonary lobe on a work surface made of waterproof paper with gloves, both sterile (Ansell, Cowansville Canada). With as limited contact with the organ as possible, the tumor was located, the margins were measured, and the tissue subsampled using sterile stainless-steel instruments (bleach 1.6%/10 min + dry autoclave cycle [121 °C, 15 psi, 45 min, 15 min cooling]). A new set of instruments was used for each type of tissue sampled and each patient to avoid carryover. A full transversal slice of 3 mm from the cancerous mass was collected and placed in a pre-weighted sterile 50 mL conical tube (Sarstedt, Newton, USA). A healthy tissue sample of ~8 cm³ from the same pulmonary lobe was also taken 5 cm away from the cancerous lesion in the matching pulmonary tiers (proximal, median, distal), measured from the origin of the lobe bronchus (Supplementary Fig. 4). A negative control, which consisted of an empty 50 mL conical tube, was handled and opened exactly the same way as the ones containing tissue. For each patient, the three tubes were flash frozen in liquid nitrogen (−196 °C) and kept in a −80 °C freezer until further use. There was a maximal interval of 30 min between tissue collection in the operating room, processing, and storage for each specimen.

**DNA extraction process**. Since it is made of 5–10% of elastin and 10–20% of collagen[55], the pulmonary matrix is highly elastic and difficult to break down. Therefore, an enzymatic and mechanical tissue disruption steps were performed.

Each patient's samples were treated in a separate batch. The tubes were thawed at 4 °C for 90 min and weighted. The pulmonary tissue was disrupted using the Liberase™ TM enzyme cocktail (Roche, Bâle, Switzerland). The lyophilized enzyme cocktail was reconstituted according to the manufacturer specifications and diluted

to 0.1 mg/L with molecular grade PBS (Thermo Scientific, Waltham, USA). Three milliliters of the 0.1 mg/L solution were used per gram of tissue. The average amount of liquid corresponding to both tissue samples was added to the control tube. The three tubes were placed in a 37 °C rotation incubator (200 r.p.m.) for 75 min.

The mechanical homogenization was achieved under a biosafety level II cabinet using a Fisherbrand™ 150 homogenizer with plastic probes (Thermo Fisher Scientific, Waltham, USA). The tissues were crushed until a uniform suspension was obtained. The controls were also agitated. A different sterilized probe (bleach 1.6%/10 min + dry autoclave cycle) was used for each tissue or control.

Three commercial DNA extraction kits were tested: QIAamp® DNA Blood Maxi Kit (Blood kit) (QIAGEN, Hilden, Germany), DNeasy® PowerLyzer® Microbial Kit (Microbial kit) (QIAGEN, Hilden, Germany), DNeasy® PowerLyzer® PowerSoil Kit (Powersoil kit) (QIAGEN, Hilden, Germany). The Blood kit was used as recommended by the manufacturer with the following modifications: 1.5 mL of the homogenous tissue suspension or control were combined with 8.5 mL of molecular grade PBS before following the manufacturer spin protocol for more than 5 mL of blood; 500 μL of QIAGEN proteases was introduced before the addition of the AL buffer. To avoid clogging the purification column, the tubes were centrifuged $3220 \times g$ for 2 min after the 70 °C/10 min incubation period. The supernatant was transferred to a new tube and, the pellet, discarded. An additional wash with the buffer AW2 of 5 mL was also performed. The column was eluted twice with the same 600 μL of AE buffer. The Microbial and Powersoil kits were used as described by the manufacturer. The disruption step was performed using a mixer mill at 30 beats/s for 20 min (Retsch, MM301, France). One-hundred microliters or 250 μL of homogenates and 50 μL or 100 μL of EB buffer for elution were used for the microbial and Powersoil kits, respectively. The DNA yield and purity were immediately measured by spectrometry. The eluates were subdivided in multiple Eppendorf tubes and kept at −80 °C until further use. The most appropriate extraction technique was selected based on the DNA yield, the purity of DNA, the observed alpha diversity, and the quantity of contaminants introduced.

**Spectrophotometric quantification of DNA**. The DNA yield and purity analyses were performed using the spectrometer NanoDrop 2000 (Thermo scientific, Waltham, Massachusetts, USA) at 260 nm, 280 nm, and 230 nm. Appropriate dilutions were performed when necessary. The quantity of extracted DNA by amount of tissue treated was calculated by multiplying the concentration of DNA in the eluate by the volume of eluate and divided by the corresponding amount of tissue treated. Paired Wilcoxon tests were performed with a significance threshold of 0.05.

**Detection of spiked bacterial mock community**. A whole-cell community and genomic community mock communities (MSA-1002, MSA-2002, ATCC, Masanass, United States of America) were used. They were both composed of 5% of either live cells or genomes of 20 bacteria (Supplementary Table 3). Healthy and cancerous tissue from two different patients were obtained the sampling procedure described above and were pooled together. The extractions were performed as described with the three different commercial kits. The tissue homogenate was either extracted alone or with a spike of $2.5 \times 10^3$ bacteria/g of tissue bacteria from the whole-cell mock community, in the tissue homogenate, before extraction. The genomic community was either sequenced on its own or spike in the human tissue extraction eluate. The average genome length of every species it includes was used to estimate a number of genomics copies. We estimate that $1.25 \times 10^3$ genomes were spiked in genomic samples. The whole-cell community was also extracted alone at the same concentration than the spiked samples (diluted in PBS). Five extracted samples in triplicates (15 samples) for each extraction kit were sent to the sequencing platform: extracted tissue DNA, extracted tissue DNA + genomic mock community, tissue + whole-cell mock community simultaneously extracted, extracted whole-cell mock community DNA, and extraction control. The genomic mock community was sequenced by itself accounting for the sample dilution. Supplementary Fig. 1 presents the experimental design of this portion of the article.

To verify the detection of the spiked bacteria, OTUs with the same taxonomic identification were agglomerated at the genus rank. Percentages of detection were computed according to the expected number of distinct bacterial genus (20). The percentages were averaged from the three replicates.

**Sequencing**. The samples were sequenced on the Illumina MiSeq 2X300 platform (IBIS, Université Laval, Quebec City, Canada) using the sequence-specific regions described by Klindworth et al. in a long oligo PCR approach[45]. The oligonucleotide sequences used to target the 16S rRNA gene V3–V4 regions are available in Supplementary Table 2. To avoid inhibition when performing library construction, the high abundance of human DNA in the extracted samples should be considered to perform appropriate dilutions.

**Data analysis and selection of techniques**. Sequences cleaning, clustering in OTUs, and taxonomical classifications were performed using Mothur's SOP version 1.40.5[43,56,57] and the SILVA reference database, release 132[58–60]. Only the

sequences between the lengths of 420 and 500 were kept in the analysis. Additional diversity analysis we conducted using RStudio[61]. The 16S data were imported and manipulated using the phyloseq package, version 1.26.1[62]. Plots were created using the packages ggplot2 version 3.2.0 and ggpubr 0.2.1[63,64]. Alpha diversity analyses were performed with the vegan, version 2.5-5, and microbiome R packages, version 1.4.2[65,66]. The Shannon diversity index was used to evaluate the alpha diversity of the samples. Double-sided paired $t$-tests were performed. The significance threshold is set at 0.05.

**Bioinformatics management of controls**. The OTUs present in the controls were removed from the OTU list of both cancerous and healthy samples for the same patient with a homemade R script[67]. Only the OTUs with a relative read count (number of reads from the OTU/total number of reads of the sample) in both tissue samples bigger than 1000× the relative count for the same OTU in the control were kept.

**Statistics and reproducibility**. No statistics were performed on the technical replicates while detecting spiked mock-communities in lung tissues. Statistics were only derived while testing the efficiency of DNA extraction. Five pairs of tissues (cancerous and healthy) from five distinct patients ($n = 5$) were processed with the three different extraction methods ($n = 30$). Paired Wilcoxon tests were performed with a significance threshold of 0.05 to compare DNA yield and purity and association between controls and tissue samples. Paired $t$-tests were performed for alpha diversity comparisons at a same significance threshold.

To assess the disparity between controls and samples depending on the extraction technique used, the percentage of remaining OTUs and Pearson's correlation were computed. The percentage of remaining OTUs was calculated by dividing the observed number of OTUs after the removal step by the observed number prior to removal. The Pearson correlation statistics were performed on the relative counts of samples and controls from the same patient before removal using the metagenomeSeq package version 1.24.1[68]. The normality of the data distribution was controlled using the Shapiro-Wilk test provided by the stats package, version 3.5.2[69]. Accordingly, paired Wilcoxon tests acknowledging for the patient and extraction technic, were performed with ggpubr, version 0.2.1[63].

In order to assess the microbial contaminant burden of the method, the bacterial profiles of the incorporated controlswere analyzed. The Bray-Curtis distance, taking into account the abundance of reads (weighted) or not (unweighted), and analyses of variance (ADONIS) at 5000 permutations was performed with the Vegan package, version 2.5-5[65]. Venn diagrams were obtained using ampvis2 package, version 2.5.8[70]. The singletons were removed before using ampvis2, only OTUs with relative abundance of 0.001% were considered and the frequency cut-off was placed at 40% to allow only OTUs found in at least two of five controls for a same extraction kit pipeline to be considered "Core". OTUs identified as "Core" were agglomerate by genus and their abundance by control and extraction kit were plotted.

**Reporting summary**. Further information on research design is available in the Nature Research Reporting Summary linked to this article.

## Data availability
The data are publicly available on the Sequence Read Archive (SRA) of the National Center for Biotechnology Information (NCBI). Accession number: PRJNA632856. The raw sequencing data as well as the DNA quantification and purity assessment data used to produce the plots and analyses are available. Supplementary Data 1 details the name and origin of the samples. The underlying data of the figures found in this article are available in Supplementary Data 2.

## Code availability
The R script used to remove the contaminants found in the experimental controls is publicly available on GitHub[67].

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

## Acknowledgements

This project was supported by the Institut Universitaire de Cardiologie et de Pneumologie de Quebec (IUCPQ), in association with the Fonds sur les maladies respiratoires J.-D.-Bégin-P.-H.-Lavoie, and the Quebec Respiratory Health Network (QRHN). N.D.L. was awarded two master's degree scholarships from the Natural Sciences and Engineering Research Council of Canada (NSERC) and the Fonds de Recherche du Québec – Santé (FRQS). He was also the recipient of short internship scholarships from the NSERC and QRHN. We thank the staff of the QRHN biobank - site IUCPQ, particularly Marie-Christine Allard and Marie-Ève Côté, the surgeons and pathologists for their involvement in the sampling campaign, as well as the graduate students of Duchaine's laboratory for their support with extraction protocols and instruments management. C.D. is holder of Tier-1 Canada Research Chair on Bioaerosols.

## Author contributions

N.D.L., M.V., C.R., P.J. and C.D. contributed to the study design. C.R and P.J. coordinated the clinical staff, patient recruitment, and tissue collection. N.D.L. was responsible for sample processing, data analysis, bioinformatics design, figure production, and writing and editing the manuscript. M.V., C.R., P.J. and C.D. revised and approved the final version.

## Competing interests

The authors declare no competing interests.
