## [Peer Review File · Communications Biology]

Reviewers' comments:

Reviewer #1 (Remarks to the Author):

The authors present a 16S rRNA gene amplicon sequencing based analysis workflow for pulmonary microbiota, from patient recruitment to data analysis, specifically in the context of lung cancer. Their particular focus is on limiting the amount of potential contaminating microbes originating from the workflow. The samples explored include five tumor samples, five matching healthy tissue samples, negative controls without added DNA, as well as a number of technical control samples with a mock microbial community.

Controlling for contamination in microbiome studies of low-biomass samples, such as lung tissue samples, is an important issue. Overall, this manuscript describes a solid workflow for tackling this problem, and presents several method comparisons that would be useful to other researchers working with similar sample material. I particularly commend the approach of having a negative control sample tube for each pair of tissue samples which goes through all the same handling as the actual samples. However, the manuscript needs some additional clarifications of details to be easier to read and interpret, and there should be some further testing related to the bioinformatic contaminant trimming.

Specific comments:

Major

1. Additional clarification is needed for what samples are actually used and presented in each phase of the study. Based on the Materials and Methods, the study setup is as follows:

1) a set of samples for technical testing, based on 2 pooled patient samples with spike-in mock community material: 5 sample configurations (with or without spike, genomic or whole-cell, etc) x 3 extraction kits -- 15 samples sequenced in total (plus additionally the genomic mock community sequenced in three dilutions - but this doesn't seem to be mentioned later, and it's not clear to me why this is even listed)

2) a set of real-life samples: 5 patients x 3 sample types (healthy/tumor/control) x 3 extraction kits -- 30 tissue samples + 15 controls sequenced, in total

I had to read the Methods several times to figure this out. I suggest adding a figure or a table that clearly lists and describes all the samples sequenced in the study and what they were used for. Figure S1/Figure 6 are great and very clear figures, but they only address the real-life samples, and the long section of text regarding the mock spike-in samples in the Methods rather confounds things.

2. Partly related to the above, Figure 3 seems to have a different number of points per kit - at a quick count, there are 5 points for Blood (which is what I would expect based on the Methods), but 9 for Microbial and Powersoil. The corresponding section of the methods ("Methodological Core Microbiota") speaks of 15 samples in relation to the OTU abundance/frequency cutoff. Why are there more than 5 Microbial and Powersoil samples in Fig 3?

3. Another detail that further adds to the confusion on numbers and types of samples: the spike-in

mock community results are not presented at all in the Results section, but only in the Discussion (where the text on this topic seems like it should be in Results). Why are there no figures/tables to show how well the known microbes in the mock community were recovered from samples (and were there differences in this whether they were handled "plain" or mixed with tissue samples)? Also, there was a setup of different ways of doing the spike-in (genomic or whole-cell), but there do not seem to be results that compare these?

4. The authors' approach to choosing which OTUs to remove as likely contaminants is a simple relative abundance ratio cutoff, applied separately for each triplet of samples (healthy tissue/tumor/negative control). I'm not convinced that this is the best solution. Since different alternatives were explored for other steps (e.g. homogenization approaches, extraction kits), I would like to see alternative options tested for the contaminant detection as well. Particularly, I suggest testing how the authors' trimming cutoff compares to decontam, an R package specifically written for contaminant detection in microbiome samples:

Davis, N.M., Proctor, D.M., Holmes, S.P. et al. Simple statistical identification and removal of contaminant sequences in marker-gene and metagenomics data. *Microbiome* 6, 226 (2018). <https://doi.org/10.1186/s40168-018-0605-2>

Minor

1. "16S" is an incomplete expression, and should not be used alone (e.g. line 189 "16S data", line 485 "16S amplicon sequencing") - the proper expression is be "16S rRNA gene", e.g. "16S rRNA gene amplicon sequencing".

2. What reference databases were used in mothur?

3. Figure 2 could show sample type (healthy/tumor) as shapes.

4. Lines 381-383, the statement "We thus hypothesize that the yields obtained in this study represents the lowest yield that can be obtained from the process." -- This does not make much sense to me; the study was not set up to estimate the lowest limit of detecting microbes from samples, and there is no clear reason to hypothesize anything about that based on the results.

5. Line 474: The start of this line seems to be an incomplete partial sentence; it's not stated what these p and R2 values refer to.

Reviewer #2 (Remarks to the Author):

The authors propose a protocol for sequencing and analyzing microbial DNA from lung tissues, paving the way for future studies on the connections between lung microbiomes and disease. They test three DNA extraction kits and find one preferable for this particular type of low microbial burden environment.

They evaluate the protocol by sequencing five patients' healthy and cancerous lung tissue with known bacterial mock communities added. The communities are mostly recovered and distinguishable from methodological controls.

The idea to include a control combining the biases/contamination from every step of the pipeline is useful and should reduce the need for step-specific controls. The idea to use the relative abundance of OTUs in samples vs. the control sounds plausible but may need further systematic validation. The authors may want to point to some more recent work in the space of bias correction and metagenomic analysis such as: "Consistent and correctable bias in metagenomic sequencing experiments" by McLaren et al. eLife (2019).

Figure 5A shows Xanthobacterecae as highly abundant in all samples, any explanation for this? Figure 5 needs a more informative caption, currently it is confusing to follow to compare A vs. B.

The discussion of limitations of this work is comprehensive and outlines possibilities for future studies. The work has its limitations but is a much needed calibration study for the particular tissue type and amplicon-based microbiome analysis. Indeed it would be interesting to see this evaluation expanded to metagenomic and metatranscriptomic sequencing.

The manuscript is overall well written but requires some proofreading.

Some specific comments:

- The title could be changed to reflect the fact that an entire pipeline is proposed, perhaps "protocol" instead of "method"
- It was somewhat difficult to parse how many replicates, controls, tumor vs. healthy samples were sequenced. There are 75 samples in the corresponding BioProject, please list them in a supplementary file with the annotation of patient number and sample type, replicate number, corresponding to the manuscript
- In the abstract please mention this is on 16S amplicon sequencing
- Line 84, explain IUCPQ
- When calling results significant, please ensure significance threshold is mentioned in Methods
- Line 205, this section is somewhat confusing, an example would be good to include
- Line 256, please start by mentioning what is compared
- Figure 4, what are the numbers presented, and the numbers in parentheses?

Reviewer #3 (Remarks to the Author):

Dumont-Leblond et al. should be commended for their attempt to establish a comprehensive methodological pipeline for the study of lung microbiota. Lung was believed to be sterile in the past, which was mainly due to the technical difficulties of culturing bacteria and low bacteria burden in respiratory tract. Recent sequencing-based studies have shown that lung bears unique microbiota, and there are strong correlations between lung microbiota and the pathogenesis of multiple diseases. However, it is still very technical challenging to exam lung microbiota. Dumont- Leblond proposed a methodological pipeline to standardize lung microbiota testing from tissue sampling to downstream analysis. The study systemically compared three different bacteria DNA extraction kits as well as other digestion methods. It is nicely written and the experimental design is clear. However, the small sample size of this study (five patients in total) is still a major concern especially given the technical challenges of measuring lung microbiota. This may explain why many results show trends but lack statistical significance. It will strengthen the conclusion by recruiting more

patients.

Rebuttal

Manuscript identification number

COMMSBIO-20-1571-T

Original title

Development of a robust method for the characterization of the pulmonary microbiota

Authors

Nathan Dumont-Leblond, Marc Veillette, Christine Racine, Philippe Joubert, Caroline Duchaine

The authors are very thankful for the thorough revision and insightful comments provided for our manuscript. We hope that this revised version as well as the response to the reviewers will meet the expectations of your journal.

Reviewer 1

1. Additional clarification is needed for what samples are actually used and presented in each phase of the study. Based on the Materials and Methods, the study setup is as follows:

- a. A set of samples for technical testing, based on 2 pooled patient samples with spike-in mock community material: 5 sample configurations (with or without spike, genomic or whole-cell, etc) x 3 extraction kits -- 15 samples sequenced in total (plus additionally the genomic mock community sequenced in three dilutions - but this doesn't seem to be mentioned later, and it's not clear to me why this is even listed)**
- b. a set of real-life samples: 5 patients x 3 sample types (healthy/tumor/control) x 3 extraction kits -- 30 tissue samples + 15 controls sequenced, in total I had to read the Methods several times to figure this out. I suggest adding a figure or a table that clearly lists and describes all the samples sequenced in the study and what they were used for. Figure S1/Figure 6 are great and very clear figures, but they only address the real-life samples, and the long section of text regarding the mock spike-in samples in the Methods rather confounds things.**

A figure was added to the supplementary materials in other to clarify the different samples created and analyzed (see below). A mention has been added to the text referring to this figure as well.

Lines 176-178 : A detailed figure presenting the experimental design of this portion of the article is available in the supplementary materials (Supplementary Materials, figure 2).

The reference to the different dilutions has been removed as they do not provide important information, are not presented in the results, and add confusion (line 201).

Figure 2 : Detailed experimental design using microbial mock-communities. Five types of samples or controls are shown. 1. Whole-cell community added to the enzymatic cocktail 2. Whole-cell community added to homogenized lung tissue, 3. Lung tissue extracted and sequenced by itself, 4. Homogenized lung tissue extract spiked with a genomic mock community, 5. Experimental control of enzymatic cocktail, 6. Genomic mock community sequenced by itself

2. **Partly related to the above, Figure 3 seems to have a different number of points per kit - at a quick count, there are 5 points for Blood (which is what I would expect based on the Methods), but 9 for Microbial and Powersoil. The corresponding section of the methods ("Methodological Core Microbiota") speaks of 15 samples in relation to the OTU abundance/frequency cutoff. Why are there more than 5 Microbial and Powersoil samples in Fig 3?**

The figure was modified to only include the 5 controls mentioned in the Methods. The additional points were from initial controls that were used to test the technique but that were not included in the manuscript to avoid confusion. We would like to thank the reviewer for pointing out this error. We also standardized the axis to standardize the two parts of the figure. The statistical tests were modified accordingly. These modifications do not change our conclusions. Please see below for more information.

3. **Another detail that further adds to the confusion on numbers and types of samples: the spike-in mock community results are not presented at all in the Results section, but only in the Discussion (where the text on this topic seems like it should be in Results). Why are there no figures/tables to show how well the known microbes in the mock community were recovered from samples (and were there differences in this whether they were handled "plain" or mixed with tissue samples)? Also, there was a setup of different ways of doing the spike-in (genomic or whole-cell), but there do not seem to be results that compare these?**

A section reporting the results of the mock communities has been added to the results section the manuscript. The report of in-depth comparisons between the different mock communities is beyond the scope of this article as it is already lengthy and encompasses numerous methods. We added a note in the methods limitation section to acknowledge our limited consideration for proportion bias.

Lines 228-239:

Detection of Spiked Bacterial Community

Whole-cell bacterial community was efficiently detected in the spiked homogenized tissues. The pipeline version using the Blood, the Microbial and the Powersoil kits detected 90,7%, 100% and 88,9% of the bacteria genera added, respectively. *Cutibacterium acnes* was not detected in any of the three samples and *Bacteroides vulgatus* was detected in only one of the three replicates using the Blood kit. *Bifidobacterium adolescentis* was detected once and *Deinococcus radiodurans*, *Clostridium beijerinckii*, *Helicobacter pylori*, *Lactobacillus gasseri* were found in 2 of the replicates using the Powersoil kit. In addition, non-spiked tissue homogenates confirmed that the genera were not naturally present in the tissue before the addition of the mock community. Only the genera *Acinetobacter* and *Staphylococcus* were detected in the tissue. All the genera from the genomic mock-community (n=20) spiked in homogenized pulmonary tissue were detected by sequencing.

Lines 519-523:

Although we put great care in reducing the incorporation of bacterial contaminants in lung tissue samples, we did not specifically measure the methodological biases. This preferential recovering and detection of some bacterial member over others is still of concerns. The resources available to correct these biases are still very scarce, but McLaren et al. offer great evidence of its importance and concrete attempts (McLaren et al., 2019).

- 4. The authors' approach to choosing which OTUs to remove as likely contaminants is a simple relative abundance ratio cutoff, applied separately for each triplet of samples (healthy tissue/tumor/negative control). I'm not convinced that this is the best solution. Since different alternatives were explored for other steps (e.g. homogenization approaches, extraction kits), I would like to see alternative options tested for the contaminant detection as well. Particularly, I suggest testing how the authors' trimming cutoff compares to decontam, an R package specifically written for contaminant detection in microbiome samples: Davis, N.M., Proctor, D.M., Holmes, S.P. et al. Simple statistical identification and removal of contaminant sequences in marker-gene and metagenomics data. *Microbiome* 6, 226 (2018). <https://doi.org/10.1186/s40168-018-0605-2>**

Thank you for this very relevant comment. We looked into the decontam package the reviewer is suggesting. It incorporates two types of contaminants removal, by frequency of sequence or prevalence throughout the dataset. As expressed by Davis *et al.*, frequency-base contaminant identification is not recommended for extremely low-biomass samples. It is therefore not applicable in lung microbiota. Furthermore, the quantities of DNA measured after amplification that are required by this method cannot be precisely obtained from our samples as a large quantity of human DNA from lung cells is also present and throws off the calculations.

The second method (prevalence-based) does not present these limitations. The function the authors presented is based on a chi-squared or fisher test performed on a 2x2 absence-presence and either control/sample table. For each OTU a table is computed to compare the group of samples and controls. If the OTU is more statistically present in the sample group, it is considered as not contaminants. This method has shortcomings as it does not account for proportion of samples. Therefore, an OTU present in only one copy (that might have been wrongly classified) in a control yields the same weight as reads of this same OTU accounting for 20% (e.i. 4000 reads) of a sample reads. Furthermore, lung microbiota characterization data that we have collected since the development of this methodology do not shown strong correlation in bacterial composition of samples between patient and the absence of a define "core microbiota" as seen in gut microbiota. The absence of recurring OTUs may undermine our ability to obtain a statistical difference between samples and control which would underestimate the diversity present in samples. The nature of the method is also not compatible with our study design. As acknowledge by the authors, the contaminant removal method is more robust as the number of samples and control increase. Our study design incorporates a single control for each pair of samples from each patient. These controls account for steps that are not shared by every sample we analyzed, as least not necessarily in the same batch, and cannot be pooled to allow for the global analysis that is recommended. Performing the analysis on pairs does not seem appropriate as the sensitivity of this method is limited when few samples are compared due to the nature of the statistical test used.

To our knowledge, no other method could be compared in the case of lung microbiota and that is compatible with the study design presented in this article. On the other hand, as identified by the Pearson's correlation we performed on pairs of control and sample, the sample extracted using the Blood DNA extraction kit share very little OTUs with their controls, reducing the need for a highly robust bioinformatics contaminant removal protocol. We still consider the method we present to be appropriate, but we added a sentence on our lack of deep characterization on the robustness of our method. We also referenced the study suggested by the reviewer mentioning the concerns we expressed here.

Lines 460-463 : Some research groups tried to use a neutral community model (Venkataraman et al. 2015), additional qPCR data (Lazarevic et al. 2016), amplicon DNA yield, or prevalence algorithms (Davis et al. 2018) to assess the influence of methodological contaminants.

Lines 481-489 : We acknowledge that our contaminant management method does not have the in-depth validation of other methods, such as described by Davis *et al.* with the decontam package (Davis et al. 2018). However, it does not share its limitations regarding the lack of consideration for OTU abundance and need of high number of controls to ensure sensitivity while using prevalence-based detection. Further research focused on the development of statistical methods to detect contaminant OTUs in the cases of lung microbiota is needed. This work is to be a starting point toward methodological standardization and its modular nature makes the bioinformatic contaminant management method we proposed interchangeable once a more robust one is uncovered.

5. "16S" is an incomplete expression, and should not be used alone (e.g. line 189 "16S data", line 485 "16S amplicon sequencing") - the proper expression is be "16S rRNA gene", e.g. "16S rRNA gene amplicon sequencing".

Corrections were made to ensure proper expressions are used.

Lines 185-186: The oligonucleotide sequences used to target the 16S rRNA gene V3-V4 regions are available in supplementary materials (table 2).

Lines 430-438: The sequencing of the 16S rRNA gene amplicon was favored over a shotgun sequencing method because of the overwhelming quantity of human DNA joining bacterial genomes in the pulmonary tissue. The 16S rRNA gene is the most used marker of bacterial identification. No consensus has been reached on the selection of the 16S rRNA gene variable region (V) to sequence for human microbiota (Mizrahi-Man et al. 2013; Pollock et al. 2018).

Lines 515-517 : Furthermore, the use of a 16S rRNA gene amplicon sequencing approach rules out the possibility of identification of fungal and viral microorganisms.

Line 787 : Table 2: 16S rRNA gene V3-V4 primer sequences

6. What reference databases were used in mothur?

The SILVA reference database release 132 was used. The information and appropriate references were added to the manuscript.

Lines 190-193 : Sequences cleaning, clustering in Operational taxonomic units (OTUs,) and taxonomical classifications were performed using Mothur's SOP version 1.40.5 (Schloss et al. 2009; Kozich et al. 2013; Schloss and al. 2019) and the SILVA reference database, release 132 (Glöckner et al., 2017; Quast et al., 2013; Yilmaz et al., 2014).

7. Figure 2 could show sample type (healty/tumor) as shapes.

Shape type was modified to include the sample type in the plots. A new version of the figure 2B has now replaced the previous version as errors in the R code used to create the underlying data was uncovered. It does not change our interpretation of the data. See below for more details

8. Lines 381-383, the statement "We thus hypothesize that the yields obtained in this study represents the lowest yield that can be obtained from the process." -- This does not make much sense to me; the study was not set up to estimate the lowest limit of detecting microbes from samples, and there is no clear reason to hypothesize anything about that based on the results.

This statement was deleted from the manuscript at the demand of the reviewer. We agree that these claims were not supported enough.

9. Line 474: The start of this line seems to be an incomplete partial sentence; it's not stated what these p and R2 values refer to.

There was indeed a duplication of information. The correction was applied and the partial sentence removed.

Lines 502-505: The extraction kit had a significant impact on the composition of the bacterial profiles found in the methodological controls, taking into account reads abundance ($p=0.019$, $R^2 = 0.14374$) or not ($p=0.0184$, $R^2 = 0.09582$).

Reviewer 2

1. The authors may want to point to some more recent work in the space of bias correction and metagenomic analysis such as: "Consistent and correctable bias in metagenomic sequencing experiments" by McLaren et al. eLife (2019).

The work describes by McLaren et al. focuses on bias correction, which must not be confused with contamination management. Their discussion also includes this distinction and the fact that those two types of microbial ecology biases are both important and not exclusive.

The methodology they suggest to effectively account for taxon biases in real-life microbial community sample is using bacterial mock-community harboring expected taxa in the samples. Bacterial characterization of the pulmonary microbiota is still very scares. Therefore, we could not create and analyze a similar community. The authors acknowledged the lack of such mock-community as a limitation in their work. Furthermore, our current unpublished work on the characterization of the cancer lung microbiota does not show strong “core species” as observed in the gut microbiota which make the selection of appropriate taxa even more challenging.

We added a reference to the article pointed out by the reviewer and acknowledged the need for biases correction method applied to the lung microbiota. However, the development of such method is outside the scope of the article.

Line 519-523: Although we put great care in reducing the incorporation of bacterial contaminants in lung tissue samples, we did not specifically measure the methodological biases. This preferential recovering and detection of some bacterial member over others is still of concerns. The resources available to correct these biases are still very scarce, but McLaren *et al.* offer great evidence of its importance and concrete attempts (McLaren et al. 2019).

2. Figure 5A shows Xanthobacterecae as highly abundant in all samples, any explanation for this?

This bacterial family includes 28 species of environmental bacteria (Oren 2014). As they are found in every control we used, they are believed to be frequent bacterial contaminant in the reagents that we used.

3. Figure 5 needs a more informative caption, currently it is confusing to follow to compare A vs. B.

A new caption has replaced the previous one to increase clarity and ease of understanding according to the modification described above.

Lines 312-315 : Figure 5 : Distribution of the relative abundance and taxonomic identification of OTUs found in the controls for each version of the pipeline. **A.** Average abundances of OTUs present in at least 40% for every version of the pipeline **B.** Average abundance of OTUs present in at least 40% of the controls from one of the version of the pipeline but not the others (different DNA extraction kits).

4. The title could be changed to reflect the fact that an entire pipeline is proposed, perhaps "protocol" instead of "method"

The title was changed as suggested by the reviewer. We agree that protocol is a more appropriate term and may refer to a combination of method which is accurate for this article. Thank you for this great suggestion.

Line 3 : Development of a Robust Protocol for the Characterization of the Pulmonary Microbiota

- 5. It was somewhat difficult to parse how many replicates, controls, tumor vs. healthy samples were sequenced. There are 75 samples in the corresponding BioProject, please list them in a supplementary file with the annotation of patient number and sample type, replicate number, corresponding to the manuscript**

An additional table had been added to the supplementary material to reduce confusion while parsing the different replicates sequenced and their location in the BioProject (see manuscript since the table is very large). This table is available in the separate excel document. Reference to the detailed experimental design using the microbial mock-community figure requested by the reviewer 1 have been made to ensure ease of comprehension.

Lines 556-560 : A table in the Supplementary material details the name and origin of the samples (table 4).

We uploaded additional data to the BioProject, as some sample turned out to be missing. All the data is now available.

- 6. In the abstract please mention this is on 16S amplicon sequencing**

The mention has been added to the abstract

Lines 34-36 : Therefore, we aim to validate a complete 16S rARN gene amplicon sequencing workflow, from patient recruitment to bioinformatics, tailored to the constrains of the pulmonary environment.

- 7. Line 84, explain IUCPQ**

The initialism has been defined in the text as requested:

Lines 83-86: It is adapted to the clinical workflow in thoracic surgery and pathology at Institut Universitaire de Cardiologie et de Pneumologie de Québec (IUCPQ) in order to minimize the impact of the implementation.

- 8. When calling results significant, please ensure significance threshold is mentioned in Methods**

The significance threshold is set at 0.05. The information has been added to the appropriate Methods sections.

Line 158-159 : Paired Wilcoxon tests were performed with a significance threshold of 0.05.

Lines 200-201 : The significance threshold is set at 0.05.

- 9. Line 205, this section is somewhat confusing, an example would be good to include**

This section as now been revised. It is now believed to be more understandable.

Lines 207-210: The singletons were removed, only OTUs with relative abundance of 0.001% were considered and the frequency cut-off was placed at 40% to allow only OTUs found in at least two of five controls for a same extraction kit pipeline to be considered “Core”.

10. Line 256, please start by mentioning what is compared

A reminder of the objects and goals of this comparison was added as an opening sentence.

Lines 275-276 : In order to assess the similarity in bacterial profiles of controls and sample (same patient and DNA extraction technique), we computed Pearson’s correlation coefficient.

11. Figure 4, what are the numbers presented, and the numbers in parentheses?

Additional precision has been added to the nature of the numbers displayed and their significance.

Lines 305-310 : Figure 4 : Venn diagram of the core microbiota in the methodological controls. The numbers of OTUs with at least 0.001% of relative abundance in two of the five control samples are displayed in each part of the diagram. The numbers in parenthesis represent sum of the average relative abundance of the OTUs in the controls. Therefore, they may not add up to 100% for each category. The non-core statistic represents the total relative abundance or number of OTUs observed in at least 0.001% relative abundance but that could not be found in at least two samples of the sample version of the pipeline.

Reviewer 3

- 1. However, the small sample size of this study (five patients in total) is still a major concern especially given the technical challenges of measuring lung microbiota. This may explain why many results show trends but lack statistical significance. It will strengthen the conclusion by recruiting more patients.**

We are unable to increase the number of samples in this study for practical reasons related to sample availability and the access to the clinical pathology laboratory given the context of COVID-19 as this laboratory is designated to handle and manage specimens potentially contaminated with COVID-19. However, we have successfully applied the final protocol we presented here to 24 other patients and will be publishing these results shortly. The controls were still reliably distinguishable from the lung tissue samples, which is a good indicator of the repeatability of our protocol.

Modifications partially or not requested by the reviewers:

- 1. The abstract has been reworked to respect the 150 words limit.**

The lack of methodological standardization diminishes the validity of the results obtained and the conclusions made when studying the lung microbiota. Therefore, we aim to validate a complete 16S rARN gene amplicon sequencing workflow, from patient recruitment to bioinformatics,

tailored to the constraints of the pulmonary microbiota. We minimized the impact of contaminants and established negative controls to track and account for them at every step. Enzymatic and mechanical homogenization combined to commercially available extraction kits allowed fast and reliable extraction of bacterial DNA. The bacterial signatures of extracted cancerous and healthy human tissues from 5 patients could be distinguished from methodological controls. The DNA extraction kits used significantly impacted the bacterial composition of the controls. Our work expands our understanding of low microbial burdened environments analysis. This paper is to be a starting point towards methodological standardization and the implementation of proper sampling procedures in the study of lung microbiota.

2. The number of control samples displayed in figure 3 and used to produce figure 4 and 5 was brought down to 5 by extraction kit (15) has described in the methodology. We removed the additional control we displayed the avoid confusion. The new figures can be seen below. The results now shown are not in contradiction to the conclusion previously made and the text was modified accordingly.

Figure 3 : Principal Coordinates Analysis of the Extraction Controls by Extraction Kits Based on Weighted (A) and Unweighted Bray-Curtis Distances (B). Data ellipse were computed based on multivariate t-distribution at 95%.

Figure 4 : Venn diagram of the core microbiota in the methodological controls. The numbers of OTUs with at least 0.001% of relative abundance in two of the five control samples are displayed in each part of the diagram. The numbers in parenthesis represent sum of the average relative abundance of the OTUs in the controls. Therefore, they may not add up to 100% for each category. The non-core statistic represents the total relative abundance or number of OTUs observed in at least 0.001% relative abundance but that could not be found in at least two samples of the sample version of the pipeline.

Line 293 : The number of OTUs was modified from 29 to 10

Lines 295-298 : The taxa Burkholderia-Caballeriona-Paraburkhloderia, Pelomonas, Sphingomonas, Streptococcus and Tropheryma were removed. The number of different OTUs was always modified from 22,74,and 47 to 35,29, and 25 according to the new figure 4.

Line 506 : The number 11 has been replaced by 10 as defined by the new computation presented in figure 4.

Figure 5 : Distribution of the relative abundance and taxonomic identification of OTUs found in the controls for each version of the pipeline. **A**. Average abundances of OTUs present in at least 40% for every version of the pipeline **B** . Average abundance of OTUs present in at least 40% of the controls from one of the versions of the pipeline but not the others (different DNA extraction kits)

3. A mistake in the code used to generate the figure 2B was uncovered. The corrected figure is now available. The trend remains the same, but significance was lost for the comparison between the Blood and Microbial kit. It does not change our overall conclusions.

Figure 2 : **A.** Pearson’s correlation coefficient between tissue samples and corresponding control. A coefficient closer to 1 or -1 illustrates a strong relationship between the bacterial profile of the sample and the control. **B.** Percentage of remaining OTUs after the removal of controls. A percentage closer to 100% shows that no OTUs present in the tissue samples had to be removed due to their presence in the controls. The boxes display the data range, quartiles and median.

Lines 285-288 : The samples extracted with the Blood kit retained a significantly larger proportion of OTUs (88.46% to 96.69%) than the Powersoil kit ($p=0.0039$). No difference could be observed between the Microbial and Powersoil or Blood kits ($p\text{-value} = 0.28, 0.11$) (Figure 2B).

4. Minor corrections

Lines 18: Additionnal affiliation was added.

4. Canada Research Chair on Bioaerosols

Lines 57-58 : Grammar corrections were applied.

A few studies aimed at analyzing **a similar** effect of lung microbiota **on** pulmonary cancer (Lee et al., 2016; Liu et al., 2018; Yan et al., 2015; Yu et al., 2016).

Lines 75-77 : The type of cancer was specified.

Recently, *Jet al.* observed a promoting effect of commensal bacteria on **lung** cancer development in mice (Jin et al. 2019).

Lines 58-60 : The type of pulmonary microbiota and methodologies were specified.

Until now, **human** pulmonary microbiota studies **integrating next-generation sequencing methodologies (NGS)** have all used different protocols for tissue collection, nucleic acids extraction and bioinformatics analyses, which limit the conclusions that can be drawn from the current literature.

Line 91-92 : The type of cancer was specified :

[...]5) Diagnosis of either **lung** adenocarcinoma or squamous cell carcinoma.

Line 93-94 : The table was specified.

The clinical details of the patients and specimens are summarized in the supplementary data **(table 1)**.

Lines 103-105 : The metal from which the instruments are made was added.

With as limited contact with the organ as possible, the tumor was located, the margins were measured, and the tissue subsampled using sterile **stainless steel** instruments (bleach 1,6%/10 min + dry autoclave cycle [121°C, 15psi, 45min, 15min cooling]).

Lines 105-106 : Precision were added to ensure proper understanding of the sentence.

A new set of instruments was used for each **type of** tissue sampled **and patient** to avoid carryover.

Line 114 : Surgery room was replaced by operating room.

Lines 129-133 : The order of the sentences was modified to facilitate reading.

The mechanical homogenization was achieved under a biosafety level II cabinet using a Fisherbrand™ 150 homogenizer with plastic probes (Thermo Fisher Scientific, Waltham, USA). The tissues were crushed until a uniform suspension was obtained. The controls were also agitated. A different sterilized probe (bleach 1.6%/10 min + dry autoclave cycle) was used for each tissue or control.

Lines 186-188 : A sentence was added.

To avoid inhibition when performing library construction, the high abundance of human DNA in the extracted samples should be considered to perform appropriate dilutions.

Line 196-197 : A duplication of package version was removed.

Plots were created using the packages ggplot2 version 3.2.0 and ggpubr and 0.2.1 (Wickham 2016; Kassambara 2019).

Line 450-452 : The sentence was not referring to the appropriate figure.

Its main features include the incorporation of a single negative control that monitors the incorporation of contaminants at every step of the experimental method (figure 6).

Line 550 : An additional acknowledgement was added.

CD is holder of Tier-1 Canada Research Chair on Bioaerosols

Line 553 : As prescribed, a competing interest statement has been added.

The authors have no competing interests to declare.

REVIEWERS' COMMENTS:

Reviewer #1 (Remarks to the Author):

The authors have addressed all my points carefully, and I find that the revisions clarifying the experimental setup are useful and well done; I particularly like the added Supplementary Figure 2. While I'm still not entirely sure if the contaminant removal approach was the best possible one, I think this is a tricky question, and the authors do have a good justification for how they do it. It will undoubtedly be a topic of further discussions in the future; for this manuscript, I think the present text where the challenges are mentioned is fine. I have no further comments or questions, and recommend this manuscript for publication.

Reviewer #2 (Remarks to the Author):

The authors have done a great job addressing the concerns of the reviewers, corrected their figures and results, and added clarifying information on the samples used.

Thank you for adding the table of sample names, it would be useful to include another column with the NCBI identifier for each sample.

Please add the possible explanation on Xanthobacteraceae abundance to the revised manuscript (it was noted as a likely contaminant in the response to reviewers).

Note a typo: In the abstract, rARN should be rRNA.

Reviewer #3 (Remarks to the Author):

We thank the authors for addressing our concerns. It is technically challenging to recruit more patients given current COVID pandemic. We look forward to authors the following study.